# The association of different presentations of maternal depression with children's socio-emotional development: A systematic review

**María Francisca Morales** \*, **Lisa-Christine Girard**, **Aigli Raouna**, **Angus MacBeth**

Department of Clinical Psychology, School of Health in Social Science, The University of Edinburgh, Edinburgh, United Kingdom

\* mariafrancisca.morales@ed.ac.uk

**Data Availability Statement:** Publicly available datasets were analyzed in this study. This data can be found at the OVID multi-field search: APA

## Abstract

Maternal depression from the perinatal period onwards is a global health priority associated with an increased likelihood of suboptimal socio-developmental outcomes in offspring. An important aspect of this association is the extent to which sustained maternal depression impacts on these outcomes. The current review synthesised the evidence on maternal depression from the perinatal period onwards and offspring internalising, externalising, and social competence outcomes. We also identified sources of methodological bias. A systematic review following PRISMA guidelines was conducted. Longitudinal studies targeting biological mothers with depressive symptomology, detailing onset, using repeated validated measures, and assessing children's outcomes between three and 12 years were included. Twenty-four studies met inclusion criteria. Findings supported the validity of different presentations of maternal depression, including consistent identification of a group of chronically depressed mothers across countries. Mothers within this group reported poorer internalising, externalising, and social competence outcomes for their offspring, with the highest levels of child problems associated with greater maternal chronicity and symptom severity. Results differed by measurement type with mothers rating poorer outcomes in comparison to teachers reports. For timing of depression, evidence was inconsistent for independent effects of antenatal or postnatal depression on child outcomes. There was substantial variability in study quality assessment. Assessing different presentations of maternal depression is essential for capturing the longitudinal associations between maternal depression and offspring outcomes to inform targets of early intervention. Chronicity, severity, and concurrent maternal depression have important implications for children's development and should be targeted in future programme planning. Further research in low- and middle-income countries is warranted.

## Introduction

The perinatal period is often challenging for women, as the transition to motherhood involves substantial changes across biological, psychological, and social domains [1]. Exposure to

PsycInfo (1806 to present); Embase Classic +Embase (1947 to present); and Ovid MEDLINE(R) and Epub Ahead of Print, In-Process & Other Non-Indexed Citations and Daily (1946 to present).

**Funding:** This work was supported by the National Agency for Research and Development (ANID)/ Scholarship Program / DOCTORADO BECAS CHILE/2019 - 72200120, for the first author, MFM. The funders had no role in study design, data collection and analysis, decision to publish, or preparation of the manuscript.

**Competing interests:** The authors have declared that no competing interests exist.

additional perinatal challenges–such as maternal depression (MD)–may increase perinatal stress, potentially representing a risk factor for the development of suboptimal developmental outcomes in their offspring [2, 3]. Although commonly occurring in around 13–15% of women, MD is a psychiatric disorder with potentially severe symptoms and complications, with onset at different stages of motherhood [4]. Maternal depressive symptoms may appear during pregnancy, in the postpartum period (up to the first year postnatally, following the World Health Organization definition), or after the postnatal period and recurrently across the offspring's childhood and adolescence [5]. Depressive symptoms during pregnancy may affect the intrauterine environment and change immature biological systems in the foetus, potentially increasing the risk for future disease [6]; and may also be associated with a woman's capacity to prepare herself cognitively and emotionally [7]. Moreover, it may affect maternal sensitivity and attunement to the baby's needs in the postpartum period with subsequent associations with the mother-child relationship [2].

Evidence has accumulated for associations between MD and children's negative outcomes, with meta-analyses showing significant small [8, 9] and medium [10] effect sizes between MD and children's behavioural and emotional outcomes from early childhood to late adolescence. More precisely, both antenatal MD and postnatal MD have been associated with increased risk of adverse developmental and psychological outcomes in children [6, 11, 12], and adolescents [13, 14]. Depressive symptoms during pregnancy may increase the risk of suboptimal birth outcomes, such as prematurity, lower birth weight, and intrauterine growth restriction at delivery, that themselves confer increased developmental risk to offspring [15–17]. Additionally, longitudinal studies using repeated measures reported that maternal depressive symptoms during the antenatal and postnatal period were associated with elevated rates of externalising [8, 12, 18, 19], internalising [18, 20], peer relationship problems and lower prosocial behaviours [21], along with difficulties reaching overall developmental milestones in childhood [22]. Longitudinal cohort studies have also shown associations between MD in the perinatal stage and adverse externalising and internalising outcomes in adolescence, such as an increase of behavioural problems [18], depression [14], and anxiety symptoms [13].

Maternal depression and its later implications for children is an up-to-date global public health priority. In 2011, the multi-stakeholder Grand Challenges in Global Mental Health–identified life-course mental health problems in general and the impact of maternal mental health on children's development as key issues [23, 24]. Additionally, systematic review evidence emphasised the importance of rigorous longitudinal studies in unpacking associations between MD and children's behavioural and emotional outcomes [25]. However, several gaps remain in understanding these longitudinal associations, particularly how different presentations of MD are associated with children's externalising, internalising, and social competence outcomes. Rather than focusing on the presence or absence of depressive symptoms, research and policy may be improved by exploring the effects of different of MD based on different groupings and trajectories.

## Differing presentations of maternal depression

Maternal depression during pregnancy and the first few years after childbirth varies in presentation between women, with some women experiencing single episode, multiple-episode, and/ or chronic recurrent depression [5]. There is evidence that women who experience transient MD in the antenatal and postpartum stage, with time-limited symptoms, subsequently return to optimal wellbeing [26]. Nevertheless, previous experience of mental health difficulties remains a significant predictor of MD [27], and antenatal MD is itself a risk factor for postnatal MD [28, 29]. Studies with repeated assessments show that whilst the majority of women with

postnatal MD show positive improvement beyond the postpartum period, approximately one-third continue to have an elevated risk of future depressive symptoms [30–32]. Therefore, for some women, perinatal depressive symptoms may predict chronic depressive disorder [26, 33, 34].

Considering the above issues of variability, it is important to account for differing presentations of MD when studying the potential impact on child development. Three factors have been suggested to be important in determining child outcomes: timing, chronicity, and severity of the maternal depressive symptoms [35, 36]. Longitudinal studies incorporating repeated measures of depression suggest chronicity, rather than the timing of MD, is a significant factor in children's behavioural problems and peer relationship difficulties [32, 36]. Indeed, for general and externalising behavioural problems, associations with postnatal MD are better accounted for by chronic depression and related risk factors, rather than by the timing of the postnatal depressive event itself [35, 37]. For internalising symptoms, there is greater evidence for an effect of postnatal MD, which remains stable after accounting for repeated maternal depressive episodes in later developmental stages and environmental risks factors [14, 38]. Additionally, more recent longitudinal studies have demonstrated that antenatal and postnatal MD may constitute different risk factors for offspring externalising and internalising outcomes [12, 14, 18, 22, 39]. Therefore, MD may also lead to different pathways for transmission of risk depending on the specific period of onset of depression, highlighting the importance of the timing of symptoms when assessing children's outcomes.

In addition to timing and chronicity of depressive symptoms, there is evidence that severity of MD is also associated with offspring difficulties [32, 40, 41]. Severity of depressive symptoms includes a continuum of problems with different levels of symptoms, including lower, moderate, and severe levels over time. However, other studies have described only limited associations between symptom severity and offspring internalising and externalising outcomes [42]. It has also been noted that the severity and chronicity of depressive symptoms are correlated with each other [30, 35]. Severe and chronic depressive traits may reflect a higher genetic predisposition for depression [37]. Furthermore, they may be associated with additional risk factors that contribute to the severity and repeated exposure to depressive symptoms, such as marital conflict, domestic violence, single parenting, and poverty [43–45]. Thus, several questions concerning the association between MD and children's outcomes remain unclear, including how distinct presentations of MD, considering timing, chronicity, and severity, are associated with offspring internalising, externalising, and social competence problems.

Accordingly, we sought to systematically review prospective studies that have measured different presentations of MD from the perinatal period onwards. Our research questions were: (i) What is the evidence for associations between different presentations of maternal depression from the perinatal period and later stages of motherhood and children's internalising, externalising, and social competence development? (ii) What is the evidence of associations between timing, chronicity, and severity of maternal depressive symptoms and children's internalising, externalising, and social competence development?

## Methodology

### Protocol

This review adhered to the Preferred Reporting Items for Systematic Reviews and Meta-Analyses (PRISMA) guidelines [46]. Details of the study protocol can be found in Morales [47] and the PRISMA checklist in S1 Checklist.

## Eligibility criteria

Inclusion criteria were as follows: longitudinal prospective cohorts targeting biological mothers with a diagnosis or symptoms of depression; using a validated measure of depression; with an onset of symptoms described in the antenatal, postnatal or first year after childbirth; repeated measurement of MD, with at least one measure of depression during the antenatal or postnatal period and one at later stages of motherhood (after the first year of childbirth); assessing the associations with children's internalising, externalising and social competence behaviours (e.g. social skills, peer relationship, prosocial behaviour) at least once from three to twelve years old; and using validated measures. Studies including a broader age range were selected if they presented subgroup analyses on children within the target range. We focused on children aged three and above given that internalising problems may be more challenging to identify in infants and toddlers due to their less developed verbal skills. Moreover, social competence may also be difficult to assess for children not yet in childcare who may have fewer opportunities to engage with peers. Exclusion criteria included cross-sectional designs, experimental designs (e.g., interventions or antidepressant treatment); qualitative studies; reviews and meta-analysis studies; case studies. Papers where MD was employed as a mediator of other variables, without a clear results section examining the association between MD and children's internalising, externalising or social competence outcomes, or with children reporting developmental disabilities were also excluded. Henceforth, we use the term maternal depression to capture both risk for depression and more severe symptoms, given the use of screening tools across a majority of included studies.

## Information sources

Three electronic databases were searched for published and unpublished literature using the OVID multi-field search: APA PsycInfo (1806 to present); Embase Classic+Embase (1947 to present); and Ovid MEDLINE(R) and Epub Ahead of Print, In-Process & Other Non-Indexed Citations and Daily (1946 to present). Additionally, grey literature was searched using the ProQuest Dissertations & Theses Global database. The search was restricted from 1992 to 2022 as Depression Disorder with postnatal onset was admitted in the ICD-10 in that year.

## Search and selection

Searches were devised in collaboration with an information specialist from The University of Edinburgh Library. Key search terms were employed in each database using a four-module approach: MD in the perinatal period, longitudinal studies, socio-emotional development (internalising, externalising or social competence), and children (e.g., 'depress* adj3 mother*', 'antenatal', 'postpartum', 'longitudinal', 'socioemotion* develop*', 'child*'). See the S1 Text for the complete search strategy. The search was accurate to February 7th, 2022. We initially identified 661 papers, which were reduced to 354 after deduplication. (See the PRISMA diagram in Fig 1). A screening pilot was first conducted by two of the researchers with the first 30 articles. Disagreements were resolved by discussion and consensus meetings. Thereafter, all 352 deduplicated study titles and abstracts were reviewed independently by these same two researchers, followed by a review of full-texts (68 studies). Inter-rater reliability after full-text review and prior to consensus resulted in Cohen's kappa of 0.692, showing good agreement [48]. A total of 24 articles met all inclusion criteria, representing 21 cohorts.

## Data extraction

A data extraction tool was created and piloted (See S2 Text). Only final models of included studies, adjusted for covariates, are discussed. Where studies did not report effect sizes, these

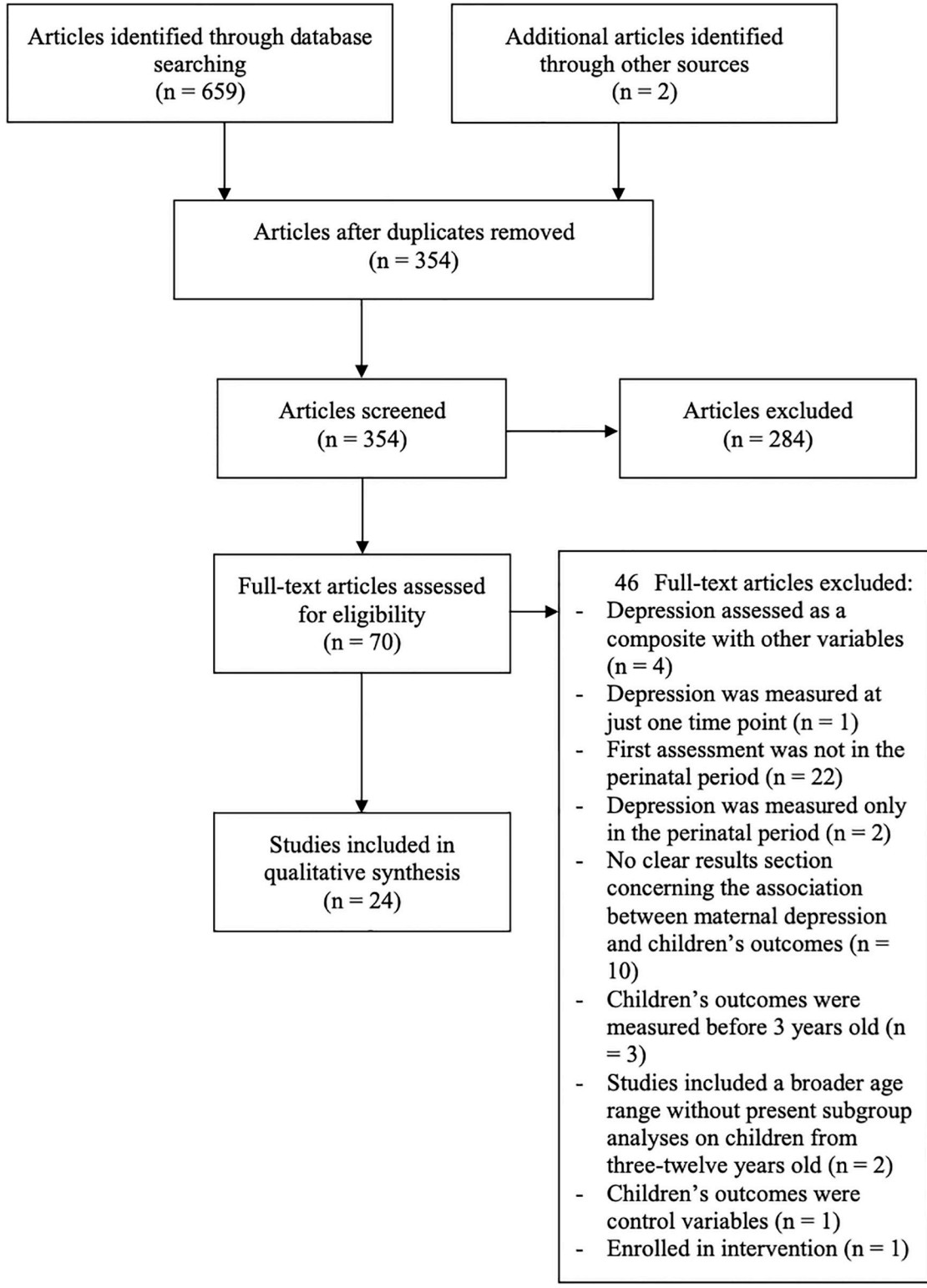

**Fig 1. PRISMA flow diagram.**

were estimated by the first author if appropriate statistics were available (means and standard deviations). Due to the lack of common model parameters of MD reported in the selected papers, a meta-analysis was not conducted. A narrative synthesis was performed instead.

## Operational definitions

For clarity and ease of interpretation, timing, chronicity, and severity were defined as follows. Timing: the period when children were exposed to maternal depressive symptoms. Timing was only assessed when papers explicitly referred to particular periods (e.g., antenatal, postnatal, concurrent at the final assessment). Chronicity: maintenance of symptoms over time, regardless of their degree of severity. Included studies defined chronic groups as either chronic, stable, persistent, increasing, or decreasing (without remitting symptoms). Severity: a continuous gradient reflecting variation of depressive symptoms, including low, moderate, and severe levels. Fluctuation of symptoms over time can be reflected by increasing, decreasing, or constant symptoms.

## Risk of bias/ Quality assessment

Methodological quality of studies was assessed using an adapted version of the Agency for Healthcare Research and Quality (AHRQ) checklist [49]. The AHRQ checklist has eleven items measuring possible sources of methodological bias. Items are scored as follows: "yes" (2 points), "partially" (1 point), "no" (0 points), "cannot tell" (0 points), or "not applicable" (N/A), with total scores ranging between 0–22 points if all items are applicable (e.g., 0–20 points if one item was not applicable). Higher scores represent lower methodological bias. A *percentage score* is calculated for each article by the *total score* (sum of points per study) divided by the *maximum score* (maximum score of applicable items) [50]. The methodological quality of all eligible papers was assessed by the first author, with a subgroup (33%) reviewed by a second independent researcher. Inter-rater reliability resulted in Cohen's kappa of 0.73, showing a good agreement rate [48]. Major discrepancies were resolved through consensus after discussion. To note, the term significant is used to denote statistical significance hereafter.

## Results

### Study characteristics

Included studies were published between 1999 and 2020. Nine of the 21 cohorts were conducted in Europe, six in North America, four in Oceania, two in Asia, two in South America, and one in Africa. Thirteen of the twenty-four studies specifically focused on modelling MD trajectories using robust statistical analyses (e.g., latent class analysis, group-based modeling, growth trajectory models) [21, 30–33, 36, 51–57], and eleven investigated maternal depressive symptoms longitudinally using their own criteria to create MD groups according to the presence or absence of symptoms [35, 58–65]. Study characteristics including timing and number of assessments are described in Table 1. Modelling approaches and results can be found in Table 2.

### Participant characteristics

Included cohorts comprised n = 54,212 mothers with sample sizes ranging from 127 [64] to 11,599 [32] at baseline. Based on reported data, mothers mean age varied from 25.4 [35] to 33.2 [56] years old at baseline. Four studies included groups of mothers without depression symptoms [55, 60, 61, 65]. As a result of attrition across waves, a total of n = 47,771 children

**Table 1. Characteristics of included studies.**

| Author, Year, Location, Language | Name of cohort | Maternal characteristics, N, Age | Offspring characteristics, N, Age | MD measurement, Type of informant | Number follow-ups, Timing of assessment | Offspring outcomes, Measurement, Type of informant | Number follow-ups, Timing of assessment |
|---|---|---|---|---|---|---|---|
| Pietikainen et al. 2020 [57], Finland, English | CHILD-SLEEP birth cohort | N = 1667(T1); N = 1421(T2); N = 1299(T3); N = 1038(T4) mean age 31.09 years old | N = 700(T5) mean age 5.71 years old | CES-D; Self-report | 32 weeks' pregnancy (T1), 3 (T2), 8(T3) months postpartum, and 2 (T4) years old | 1. Internalisng and externalising behaviours; SDQ; Parents report. | 5 (T5) years old |
| | | | | | | 2. Internalisng and externalising behaviours; Emotional/ behavioural problems subscale of FTF; Parents report | |
| Oh et al. 2020 [55], Korea, English | Panel Study on Korean Children (PSKC) | N = 1504(T4), mean age 31.1 years old | N = 1191(T9) 9 years old | K6; Self-report | One-month prior childbirth(T1), 6 (T2), and 1(T3), 2 (T4) years old | Internalisng and externalising behaviours; K-CBCL; Parent report | 4 (T5), 5(T6), 6 (T7), 7(T8) and 9(T9) years old |
| Hentges et al. 2020 [60], Canada, English | All our families (AOF) | N = 1992(T7), mean age 30.87 years old | N = 1992(T9) 5 years old | EPDS; Self-report (T1-T4); CES- D; Self-report(T5-T7) | <25 weeks gestation(T1), 34– 36 weeks gestation (T2), 4 months postpartum(T3), 1 (T4), 2(T5), 3(T6), and 5(T7) years old | Internalising and externalising behaviours; BASC-2; Mothers report | 5 (T7) years old |
| Garman et al. 2019 [52], South Africa, English | | N = 446(T4) mean age 26 at childbirth | N = 446(T4) 36 months old | EPDS; Self-report | 2(T1) weeks, 6 (T2), and 18(T3) months postpartum | 1. Internalising and externalising behaviours; CBCL; Mothers report. | 36(T4) months old |
| | | | | | | 2. Internalising and externalising behaviours; SDQ; Mothers report | |
| Maruyama et al. 2019 [21], Brazil, English | Pelotas Birth Cohort | N = 4231(T1) mean age 26.2 years old | N = 3531(T6) 11 years old | EPDS; Self-report | 3(T1), 11.9(T2), 23.9(T3), 49.5(T4) months postpartum, and 6.8(T5), 11(T6) years old | Peer relationship problems and prosocial behaviour; SDQ; Mothers report | 11(T6) years old |
| Gjerde et al. 2017 [67], Norway, English | Norwegian Mother and Child Cohort Study (MoBa) | N = 11,599 (T1) age nor reported | N = 11,599 (T6) 5 years old | SCL; Self-report | 17(T1), 30(T2) weeks' pregnancy, 6(T3) months postpartum, and 1.5(T4), 3(T5) and 5(T6) years old | Internalisng and externalising behaviours; CBCL; Mothers report | 5(T6) years old |
| Wolford et al. 2017 [66], Finland, English | Prediction and Prevention of Pre- eclampsia and Intrauterine Growth Restriction (PREDO) study | N = 1,779 (T1) mean age at delivery 31.9 years old | N = 1,779 (T15) 3–6 years old | 1. CES-D; Self- report. 2. BDI; Self-report | Biweekly during pregnancy (T1-T14), 3–6 (T15) years old | CHI; Mothers report | 3-6(T5) years old |
| Matijasevich et al. 2015 [33], Brazil, English | Pelotas Birth Cohort | N = 4231(T1) mean age 26.2 years old | N = 3585(T5) 6 years old | EPDS; Self-report | 3(T1), 11.9(T2), 23.9(T3), 49.5(T4) months postpartum, and 6.8(T5) years old | 1. Internalisng and externalising behaviours; SDQ; Mothers report. | 6(T5) years old |
| | | | | | | 2. Psychiatric diagnoses (any, internalisng, externalising); DAWBA; Structured interview by trained psychologists | |

*(Continued)*

**Table 1.** (Continued)

| Author, Year, Location, Language | Name of cohort | Maternal characteristics, N, Age | Offspring characteristics, N, Age | MD measurement, Type of informant | Number follow-ups, Timing of assessment | Offspring outcomes, Measurement, Type of informant | Number follow-ups, Timing of assessment |
|---|---|---|---|---|---|---|---|
| Ahun et al. 2018 [51], Canada, English | Québec Longitudinal Study of Child Development | N = 1537; 88 had less than 21 years old at birth, 1449 had 21 or more years old at birth | N = 1537 6, 7, 8, 10, and 12 years old | CES- D; Self-report | 5(T1) months postpartum, 1 year and 6(T2) months, 3 years and 6(T3) months, and 5(T4) years old | 1. Internalising and externalising behaviours; Preschool Behaviour Questionnaire; Mothers report | Teachers report: 6(T5), 7 (T6), 8(T7), 10 (T8), 12(9) years old Mothers report: 6(T5), 8(T7) years old |
| | | | | | | 2. Internalising and externalising behaviours; Preschool Behaviour Questionnaire; Teachers report | |
| | | | | | | 3. Internalising and externalising behaviours; Composite between measurements 1 and 2; Mothers and teachers report | |
| Kingston et al. 2018 [54], Canada, English | All our families (AOF) | N = 1983(T5) 73.1% of the sample was between 25 and 34 years old | N = 1983(T5) 3 years old | EPDS; Self-report | Before 25(T1) weeks of pregnancy, between 34 and 36 (T2) weeks of pregnancy, 4(T3) months postpartum, and 1 (T4) year old | Internalising and externalising behaviours; NLSCY Behavior Scales; Mothers report | 3(T5) years old |
| Park et al. 2018 [56], Canada, English | | N = 182(T1) mean age 33.2 years old | N = 103(T8) 6 years old | 1. HAM-D; Self-report 2. EPDS; Self-report | Second(T1) and third(T2) trimesters of pregnancy, 6(T3) weeks, 3(T4), 6 (T5), 10(T6) months postpartum, and 3 (T7), 6(T8) years old | 1. Internalising and externalising behaviours; CBCL; Mothers report 2. Child mental health symptomatology; HBQ; Mothers report | 3(T7), and 6 (T8) years old |
| Netsi et al. 2018 [32], UK, English | ALSPAC | N = 9848(T1) mean age 28.5 years old; N = 6182(T8) | N = 8419 3.5 years old | EPDS; Self-report | 2(T1), 8(T2), 21 (T3), 33(T4), 61 (T5), 73(T6), 93 (T7), 134(T8) months postpartum | Behavioural problems; Rutter Total Problems Scale; Mothers report | 3.5 years old |
| Vakrat et al. 2018 [65], Israel, English | | N = 680(T1); N (exposure) = 46(T4), N(control) = 103 (T4) mean age 38.66 years old | N = 142(T4) mean age 6.33 years old | 1. BDI; Self-report 2. Structured Clinical Interview for DSM-IV; Clinically diagnosed | 2 post-birth day (T1),6(T2), 9(T3) months postpartum, and 6 (T4) years old | Psychiatric diagnoses; DAWBA; Structured interview by clinical psychologists | 6(T4) years old |
| Giallo et al. 2015 [53], Australia, English | Maternal Health Study (MHS) | N = 1483(T1) 86.6% of the sample was between 25 and 39 years old | N = 1085(T8) 4 years old | EPDS; Self-report | 10-24(T1), 30-32 (T2) weeks pregnancy, 3(T3), 6(T4), 9(T5), 12 (T6), 18(T7) months postpartum, and 4 (T8) years old | Emotional and behavioural symptoms; SDQ; Mothers report | 4(T8) years old |

*(Continued)*

**Table 1.** (Continued)

| Author, Year, Location, Language | Name of cohort | Maternal characteristics, N, Age | Offspring characteristics, N, Age | MD measurement, Type of informant | Number follow-ups, Timing of assessment | Offspring outcomes, Measurement, Type of informant | Number follow-ups, Timing of assessment |
|---|---|---|---|---|---|---|---|
| Van Der Waerden et al. 2015 [36], France, English | EDEN | N = 1899(T1) mean age 30.13 years old; N = 1183(T7) | N = 1183(T7) 5 years old | 1. CES-D; Self-report 2. EPDS; Self-report | 24(T1) weeks pregnancy, 4(T2), 8(T3), 12(T4), months postpartum, and 3 (T5), 4(T6), 5(T7) years old | Internalisng and externalising behaviours; SDQ; Mothers report | 5(T5) years old |
| Agnafors et al. 2013 [58], Sweden, English | | N = 1723(T1) mean age 28.2; N = 893 (T2) | N = 893(T2) 12 years old | 1. EPDS; Self-report 2. HSCL-25; Self-report | 3(T1) months postpartum, and 12(T2) years old | Internalising and externalising behaviours; CBCL; Mothers report | 12(T2) years old |
| Cents et al. 2013 [31], Netherlands, English | Generation R | N = 4167(T4) mean age 31.4 | N = 4167(T4) 36 months old | BSI; Self-report | 20(T1) weeks pregnancy, 2(T2), 6(T3), 36(T4) months postpartum | Internalising and externalising behaviours; CBCL; Mothers and fathers report | 36(T4) months old |
| Fihrer et al. 2009 [59], Australia, English | | N = 127(T1); N = 75 (T5) | N = 75(T5) 6–8 years old | 1. CES-D; Self-report 2. CIDI; Clinically diagnosed | 4(T1), 12(T2), 15 (T3) months postpartum, and 4 (T4), 6-8(T5) years old | 1. Internalising and externalising behaviours; CBCL; Mothers, Fathers and Teachers report | 6-8(T5) years old |
| | | | | | | 2. Internalising and externalising behaviours; ADIS-P; Structured interview to the mother by trained postgraduate-student clinicians | |
| Campbell et al. 2007 [30], USA, English | NICHD Study | N = 1261(T1) mean age 28.27 years old | N = 1025(T7) 7 years old | CES-D; Self-report | 1(T1), 6(T2), 15 (T3), 24(T4), 36 (T5), 54(T6) months postpartum, and 7 (T7) years old | 1. Internalising and externalising behaviours; CBCL; Mothers and Teachers report | 7(T7) years old |
| | | | | | | 2. Social skills; Social Skills Rating System; Mothers and Teachers report | |
| | | | | | | 3. Children's self-reliance and positive affect; Observations of child behaviour in school; blind trained observer | |
| NICHD 1999 [63], USA, English | NICHD Study | N = 1215(T1) mean age 28.27 years old | N = 1215(T5) 36 months old | CES-D; Self-report | 1(T1), 6(T2), 15 (T3), 24(T4), 36 (T5) months postpartum | 1. Internalising and externalising behaviours; CBCL; Mothers report | 36(T5) months old |
| | | | | | | 2. Social competence; ASBI; Mothers report | |
| Josefsson et al. 2007 [61], Sweden, English | | N = 675 (index group = 221; control group = 454) mean age index group 33.5, mean age in control group 33.3 | N = 675(T3) 4 years od | EPDS; Self-report | 6-8(T1) weeks, 6 (T2) months postpartum, and 4 (T3) years old | Child's general behaviour; PBCL; Mothers report | 4(T3) years old |

*(Continued)*

**Table 1.** (Continued)

| Author, Year, Location, Language | Name of cohort | Maternal characteristics, N, Age | Offspring characteristics, N, Age | MD measurement, Type of informant | Number follow-ups, Timing of assessment | Offspring outcomes, Measurement, Type of informant | Number follow-ups, Timing of assessment |
|---|---|---|---|---|---|---|---|
| Trapolini et al. 2007 [64], Australia, English | | $N = 127$(T1); $N = 92$ (T4) mean age 34.9 | $N = 92$(T4) mean age 50.8 | 1. CES- D; Self-report 2. CIDI; Clinically diagnosed | 4(T1), 12(T2), 15 (T3) months postpartum, and 4 (T4) years old | Internalising and externalising behaviours; CBCL; Parents and Teachers report | 4(T4) years old |
| Luoma et al. 2001 [62], Finland, English | | $N = 349$(T1); $N = 147$(T5) mean age 34.7 | $N = 147$(T5) 6 years old | EPDS; Self-report | Late pregnancy (T1), first week (T2), 2(T3), 6(T4) months postpartum, and 6 (T5) years old | 1. Internalising and externalising behaviours; CBCL; Mothers report | 6(T5) years old |
| | | | | | | 2. Internalising and externalising behaviours; TRFs; Teachers report | |
| Brennan et al. 2000 [35], Australia, English | | $N = 4953$ mean age 25.4(T2) | $N = 4953$(T4) 5 years old | DSSI; Self-report | Pregnancy(T1), 3-4(T2) post-birth day, 6(T3) months postpartum, and 5 (T4) years old | Internalising and externalising behaviours; CBCL; Mothers report | 5(T4) years old |

**Notes**: ADIS-P: Anxiety Disorders Interview Schedule—Parent version, ASBI: Adaptive Social Behaviour Inventory, BASC-2: Behavior Assessment System for Children: Second Edition, BDI: Beck Depression Inventory, BSI: Brief Symptoms Inventory, CBCL: Child Behaviour Checklist, CES-D: Center for Epidemiologic Studies-Depression, CHI: Conners' Hyperactivity Index, CIDI: Composite International Diagnostic Interview, CNSIE: Nowick-Strickland Internal-External Scale, CSL: Short form of the Symptom Checklist, DAWBA: Development and Well-Being Assessment, DSSI: The Delusions-Symptoms-States Inventory, EPDS: Edinburgh Postnatal Depression Scale, FTF: Five to Fifteen, HAM-D: Hamilton Rating Scale for Depression, HBQ: Health and Behavior Questionnaire, HSCL-25: Hopkins Symptoms Checklist, K-CBCL: Korean-Child Behavior Checklist, K6: Kessler's six-question Short-form Scale self-reported questionnaire, N: Number, NLSC: National Longitudinal Survey of Children and Youth, PBCL: Pre-School Behaviour Checklist, SDQ: Strengths and Difficulties Questionnaire, T1: Time 1, T2: Time 2, T3: Time 3, T4: Time 4, T5: Time 5, T6: Time 6, T7: Time 7, T8: Time 8, TRF: Teacher Report Form.

were identified at the final assessment stage, with sample sizes ranging from 75 [59] to 11,599 [32]. See Table 1.

## Maternal depression different presentations

Several MD groups were identified longitudinally, ranging from two [51, 62, 65] to seven identified groups [32]. All but three studies [56, 59, 62] identified i) a no or low depressive symptom group (66.5% of mothers on average); and ii) a high chronic depressive symptom group (8.8% of mothers on average). In addition, 19 papers reported transient groups including, increasing, and decreasing symptoms, moderate or subclinical symptoms, and episodic depressive symptoms (e.g., pregnancy, postpartum, current). A full description of MD groups and percentage of mothers in each category are presented in Table 2.

## Internalising behaviours

Seventeen studies examined children's internalising behaviours. Using maternal reports (sixteen studies), children's internalising behaviours were associated with MD groups in all but one study [52]. In general, mothers in the chronic and more severe groups of depressive symptoms reported offspring with significantly higher internalising behaviours than mothers in low or no depressive symptoms groups. The effect size of association was large in two studies

**Table 2. Results of MD groups and significant associations with offspring outcomes.**

| Author, Year | Modelling approach, MD groups | Total (internalising and externalising) | Internalising behaviours | Externalising behaviours | Social competence | Covariates adjusted in the final model |
|---|---|---|---|---|---|---|
| Pietikainen et al. 2020 [57] | Latent profile analysis; 1. stable low, 2. stable intermediate, 3. stable high | SDQ: $F = 16.46$, $p < .001$, Partial Eta squared = .047; Group comparisons: 1 vs 2***; 1 vs 3***; FTF: $F = 15.11$, $p < .001$, Partial Eta squared = .043; Group comparisons: 1 vs 2***; 1 vs 3** | SDQ: $F = 8.66$, $p < .001$, Partial Eta squared = .025; Group comparisons: 1 vs 2***; FTF: $F = 15.30$, $p < .001$, Partial Eta squared = .044; Group comparisons: 1 vs 2***; 1 vs 3* | SDQ: $F = 11.57$, $p < .001$, Partial Eta squared = .034; Group comparisons: 1 vs 2**; 1 vs 3***; FTF: $F = 10.13$, $p < .001$, Partial Eta squared = .029; Group comparisons: 1 vs 2*; 1 vs 3** | | Maternal age, maternal education, paternal age, paternal education, maternal income, paternal income, number of previous children, child's gestational age, birthweight, gender, child's age when the questionnaire was answered |
| Oh et al. 2020 [55] | Latent profile analysis; Three depression trajectories: 1. no depression group (42.2%), 2. mild depression group (46.9%), 3. moderate depression group (10.9%) | Age 4: significant group difference, $p < .001$, Group comparisons: 1 vs 2***; 1 vs 3***; Age 5: significant group difference, $p < .001$, Group comparisons: 1 vs 2***; 1 vs 3***; Age 6: significant group difference, $p < .001$, Group comparisons: 1 vs 2***; 1 vs 3***; Age 7: significant group difference, $p < .001$, Group comparisons: 1 vs 2***; 1 vs 3***; Age 9: significant group difference, $p < .001$, Group comparisons: 1 vs 2***; 1 vs 3*** | Age 4: significant group difference, $p < .001$, Group comparisons: 1 vs 2***; 1 vs 3***; Age 5: significant group difference, $p < .001$, Group comparisons: 1 vs 2***; 1 vs 3***; Age 6: significant group difference, $p < .001$, Group comparisons: 1 vs 2***; 1 vs 3***; Age 7: significant group difference, $p < .001$, Group comparisons: 1 vs 2***; 1 vs 3***; Age 9: significant group difference, $p < .001$, Group comparisons: 1 vs 2***; 1 vs 3*** | Age 4: significant group difference, $p < .001$, Group comparisons: 1 vs 2***; 1 vs 3***; Age 5: significant group difference, $p < .001$, Group comparisons: 1 vs 2***; 1 vs 3***; Age 6: significant group difference, $p < .001$, Group comparisons: 1 vs 2***; 1 vs 3***; Age 7: significant group difference, $p < .001$, Group comparisons: 1 vs 2***; 1 vs 3***; Age 9: significant group difference, $p < .001$, Group comparisons: 1 vs 2***; 1 vs 3*** | | |
| Hentges et al. 2020 [60] | Separate groups according to presence or absence of depressive symptoms; Five groups: 1. no depressive episodes, 2. one depressive episode, 3. two depressive episodes, 4. three depressive episodes, 5. four or more depressive episodes | | $Fs(4) = 9.04–27.98$, $ps < .001$, partial etas squared = 0.02–0.07 | $Fs(4) = 9.04–27.98$, $ps < .001$, partial etas squared = 0.02–0.07 | | Child sex, maternal education, family income, maternal age, gestational age at birth, maternal health risk during pregnancy using data collected from pregnancy medical records |
| Garman et al. 2019 [52] | Latent class analysis; 1. chronic low trajectory (71.1%), 2. late postpartum trajectory (10.1%), 3. early postpartum trajectory (14.4%), 4. chronic high trajectory (4.5%) | CBCL: Standardised B(CI) significance level: 1 = (ref); 2 = non-significant; 3 = non-significant; 4 = non-significant; SDQ: Standardised B(CI) significance level: 1 = (ref); 2 = non-significant; 3 = non-significant; 4 = non-significant | CBCL: Standardised B (CI) significance level: 1 = (ref); 2 = non-significant; 3 = non-significant; 4 = non-significant; SDQ: Standardised B(CI) significance level: 1 = (ref); 2 = non-significant; 3 = non-significant; 4 = non-significant | CBCL: Standardised B (CI) significance level: 1 = (ref); 2 = non-significant; 3 = non-significant; 4 = non-significant; SDQ: Standardised B(CI) significance level: 1 = (ref); 2 = non-significant; 3 = non-significant; 4 = non-significant | | Age, education, wealth status and concurrent depressive symptoms |

*(Continued)*

**Table 2.** (Continued)

| Author, Year | Modelling approach, MD groups | Total (internalising and externalising) | Internalising behaviours | Externalising behaviours | Social competence | Covariates adjusted in the final model |
|---|---|---|---|---|---|---|
| Maruyama et al. 2019 [21] | Group-based semiparametric method; 1. low, 2. moderate-low (group 1 and 2 = 74.8%), 3. increasing (11.1%), 4. decreasing (9.2%), 5. high-chronic (4.9%) | | | | OR(CI) significance level: Peer relationship problems: 1 = (ref); 2 = 1.77(1.3 to 2.4)\*\*\*; 3 = 3.59(2.5 to 5.1)\*\*\*, 4 = 3.35 (2.3 to 4.9)\*\*\*, 5 = 4.59(2.9 to 7.1)\*\*\*; Prosocial behavior: 1 = (ref); 2 = 2.02(0.8 to 5.2)\*\*\*; 3 = 2.05(0.6 to 6.9)\*\*\*, 4 = 3.79 (1.2 to 11.7)\*\*\*, 5 = 4.51(1.3 to 15.7)\*\*\* | Family income, skin colour, maternal age, maternal schooling, marital status, planned pregnancy, parity, smoking during pregnancy, alcohol during pregnancy, depression during pregnancy, started prenatal care, type of delivery, low birthweight, gestational age, Apgar score, duration of breastfeeding, siblings, IQ, father's presence in child's life |
| Gjerde et al. 2017 [67] | Separate groups according to presence or absence of depressive symptoms; Five groups: 1. 17th gestational week, 2. 30th gestational week, 6 months postpartum, 4. concurrently | | Fixed effects estimate (CI) significance level: 1 = 0.96(0.41 to 1.50)\*\*\*; 2 = 1.42(0.84 to 2.00)\*\*\*; 3 = 2.06 (1.52 to 2.59)\*\*\*; 4 = 3.88(3.45 to 4.31)\*\*\* | Fixed effects estimate (CI) significance level: 1 = 0.56 (0.04 to 1.10)\*; 2 = 1.06 (0.50 to 1.62)\*\*\*; 3 = 1.12 (0.61 to 1.64)\*\*\*; 4 = 3.89 (3.48 to 4.29)\*\*\* | | Child age, child sex, maternal parity and education |
| Wolford et al. 2017 [66] | Latent profile analysis; 1. low, 2. high | | | OR(CI) significance level: 1 = (ref); 2 = 2.80 (2.20 to 3.57)\*\*\* | | Maternal age at delivery, antidepressant use, psychotropic medication use, smoking during pregnancy, parity, prepregnancy/chronic hypertension, type 1 diabetes, child's sex, gestational length, birthweight, family structure, maternal alcohol use and education |
| Matijasevich et al. 2015 [33] | Group-based semiparametric method; 1. low, 2. moderate-low (group 1 and 2 = 75.7%), 3. increasing (9%), 4. decreasing (9.9%), 5. high-chronic (5.4%) | Any psychiatric disorder: OR(CI) significance level: 1 = (ref); 2 = 1.9(1.4 to 2.6)\*\*\*; 3 = 3.1(2.1 to 4.6)\*\*\*, 4 = 3.3(2.3 to 4.8)\*\*\*, 5 = 8.8(5.8 to 13.6)\*\*\* | OR(CI) significance level: 1 = (ref); 2 = 1.7 (1.2 to 2.4)\*\*\*; 3 = 2.6 (1.7 to 4.1)\*\*\*, 4 = 3.1 (2.0 to 4.8)\*\*\*, 5 = 6.9 (4.3 to 11.1)\*\*\* | OR(CI) significance level: 1 = (ref); 2 = 2.1 (1.2 to 3.9)\*\*\*; 3 = 4.4 (2.2 to 8.9)\*\*\*, 4 = 3.9 (1.9 to 7.9)\*\*\*, 5 = 8.9 (4.2 to 18.7)\*\*\* | | Family income, schooling, age, marital status, skin colour, depression and smoking and alcohol during pregnancy. Gender and intermediate or intensive care hospitalization after birth |

(*Continued*)

**Table 2.** (Continued)

| Author, Year | Modelling approach, MD groups | Total (internalising and externalising) | Internalising behaviours | Externalising behaviours | Social competence | Covariates adjusted in the final model |
|---|---|---|---|---|---|---|
| Ahun et al. 2018 [51] | Group-based semiparametric method; 1. low trajectory (81.7%), 2. High trajectory (18.3%) | | OR(CI) significance level: Children with high-increasing internalising problems: 1 = (ref); 2 = 1.63 (1.03 to 2.58)*; Children with low-moderate internalising problems: 1 = (ref); 2 = 1.63 (1.06 to 2.52)*; Children with high internalising problems: 1 = (ref); 2 = 2.60 (1.55 to 4.36)* | | | Child's sex, maternal psychopathology (anxiety and antisocial behaviour), stimulation in mother–child interactions, maternal parenting practices (e.g. over protection and perception of parental impact), whether or not the mother was employed, socioeconomic status, and family functioning |
| Kingston et al. 2018 [54] | Latent class analysis; 1. minimal depressive symptoms (64.7%), 2. early postpartum depressive symptoms (10.9%), 3. subclinical depressive symptoms (18.8%), 4. high depressive symptoms (5.6%) | | OR(CI) significance level: Emotional disorder and anxiety 1 = (ref); 2 = non-significant; 3 = non-significant; 4 = non-significant; Separation anxiety: 1 = (ref); 2 = 2.16 (1.48 to 3.16)***; 3 = 1.50 (1.08 to 2.08)**; 4 = 1.89 (1.07 to 3.34)** | OR(CI) significance level: Hyperactivity and inattention 1 = (ref); 2 = 1.84 (1.24 to 2.74)**; 3 = 1.46 (1.04 to 2.06)**; 4 = 2.17 (1.22 to 3.88)**; Physical aggression: 1 = (ref); 2 = non-significant; 3 = 1.64 (1.13 to 2.36)**; 4 = non-significant | | Maternal age, educational level, family income, maternal primary language spoken at home, history of mental health disorders, symptoms of anxiety during pregnancy, gestational age, child sex, symptoms of postpartum anxiety, symptoms of depression at 3 years old, symptoms of anxiety at 3 years old, happiness in partner relationship |
| Park et al. 2018 [56] | Growth mixture modelling; 1. low (71.4%), 2. increasing (18.4%), 3. decreasing (10.2%) | Standardised B(CI) significance level: 3 years (CBCL): 1 = (ref); 2 = 10.3 (4.9 to 15.7)***; 3 = non-significant | Standardised B(CI) significance level: 3 years (CBCL): 1 = (ref); 2 = 9.1 (3.4 to 14.9)**; 3 = non-significant; 6 years (HBQ): 1 = (ref); 2 = non-significant; 3 = non-significant | Standardised B(CI) significance level: 3 years (CBCL): 1 = (ref); 2 = 9.7 (4.2 to 15.2)***; 3 = non-significant; 6 years (HBQ): 1 = (ref); 2 = non-significant; 3 = non-significant; HBQ-ADHD: 1 = (ref); 2 = non-significant; 3 = -0.4 (-0.7 to -0.1)** | | Child's sex, age, gestational age at birth, birthweight, prenatal SSRI antidepressant exposure, maternal history of depression, maternal education, and maternal minority status. Concurrent maternal depressive symptoms were included as a covariate in all models examining 6-year outcomes. |
| Netsi et al. 2018 [32] | Linear growth modelling; 1. Below threshold (91.4%), 2. Moderate but not persistent (2.6%), 3. Marked but not persistent (1.4%), 4. Severe but not persistent (2.0%), 5. Moderate persistent (1.3%), 6. Marked persistent (0.7%), 7. Severe persistent (0.6%) | | | OR(CI) significance level: Behavioural Problems: 1 = (ref), 2 = 2.22 (1.74 to 2.83)***, 3 = 1.91 (1.36 to 2.68)***, 4 = 2.39 (1.78 to 3.22)***, 5 = 3.04 (2.10 to 4.38)***, 6 = 2.84 (1.71 to 4.71)***, 7 = 4.84 (2.94 to 7.98)*** | | Maternal education |

*(Continued)*

**Table 2.** (Continued)

| Author, Year | Modelling approach, MD groups | Total (internalising and externalising) | Internalising behaviours | Externalising behaviours | Social competence | Covariates adjusted in the final model |
|---|---|---|---|---|---|---|
| Vakrat et al. 2018 [65] | Separate groups according to presence or absence of depressive symptoms; 1. chronic maternal depression, 2, control (non-depressed mothers) | $x2$ = 32.85, df = 1, p < .001; Cramer's V = 0.47 | Prevalence of anxiety disorders: $x2$ = 14.51, df = 1, p < .001; Cramer's V = 0.32 | Prevalence of oppositional defiant disorder: $x2$ = 7.97, df = 1, p = .006; Cramer's V = 0.2; Attention deficit hyperactivity disorder: $x2$ = 5.69, df = 1, p = .02; Cramer's V = 0.2 | | |
| Giallo et al. 2015 [53] | Latent class analysis; 1. minimal depressive symptoms (61.0%), 2. sub-clinical depressive symptoms (30.2%), 3. increasing and persistently high depressive symptoms (8.8%) | OR(CI) significance level: 1 = (ref); 2 = 2.51 (1.61 to 3.92)***; 3 = 2.00 (1.01 to 3.96)* | | | | Maternal age, country of birth, relationship status, highest educational attainment, employment status and hours per week, personal income, child gender, paid parental leave, whether they had subsequent children, smoked during pregnancy, or had concerns for their baby's health at 6 months postpartum |
| Van Der Waerden et al. 2015 [36] | Group-based semiparametric method; 1. no symptoms (62.0%), 2. persistent intermediate-level depressive symptoms (25.3%), 3. persistent high depressive symptoms (4.6%), 4. high symptoms in pregnancy only (3.6%), 5. high symptoms in the child's preschool period only (4.6%) | Standardised B(CI) significance level: 1 = 0.00 (ref); 2 = 1.69 (0.76 to 2.63)***; 3 = 5.55 (3.73 to 7.36)***; 4 = non-significant; 5 = 2.89 (1.32 to 4.48)*** | Standardised B(CI) significance level: Emotional symptoms: 1 = 0.00 (ref); 2 = 0.54 (0.19 to 0.89)**; 3 = 1.36 (0.68 to 2.04)***; 4 = non-significant; 5 = 0.94 (0.35 to 1.53)** | Standardised B(CI) significance level: Conduct problems: 1 = 0.00 (ref); 2 = 0.39 (0.01 to 0.76)*; 3 = 1.61 (0.88 to 2.34)***; 4 = non-significant; 5 = non-significant; Symptoms of hyperactivity/inattention: 1 = 0.00 (ref); 2 = 0.54 (0.11 to 0.97)**; 3 = 1.37 (0.54 to 2.20)***; 4 = non-significant; 5 = 0.78 (0.05 to 1.49)* | Standardised B(CI) significance level: Peer relationship problems: 1 = 0.00 (ref); 2 = non-significant; 3 = 1.21 (0.73 to 1.68)***; 4 = non-significant; 5 = 0.67 (0.25 to 1.08)**; Prosocial behaviour: 1 = 0.00 (ref); 2 = non-significant; 3 = -0.91 (-1.52 to -0.30)**; 4 = non-significant; 5 = -0.57 (-1.10 to 0.03)* | Study centre, child's sex, preterm birth, small for gestational age, duration of breastfeeding, parental separation, age mother, low income, education level of mother, number of siblings, childcare, domestic violence, paternal substance abuse, social support, maternal anxiety, history of mental health problems, maternal substance use before pregnancy, maternal antidepressant use, concurrent maternal depression at age 5 years |
| Agnafors et al. 2013 [58] | Separate groups according to presence or absence of depressive symptoms; 1. no depressive symptoms, 2. depressive symptoms at baseline, 3. depressive symptoms at follow-up, 4. depressive symptoms on both occasions | | OR(CI) significance level: 1 = (ref); 2 = non-significant; 3 = 4.97(2.74 to 9.01)***, 4 = 8.13(3.15 to 21.03)*** | OR(CI) significance level: 1 = (ref); 2 = non-significant; 3 = 3.81(2.18 to 6.67)***, 4 = 7.2(2.96 to 17.49)*** | | |

(*Continued*)

**Table 2.** (Continued)

| Author, Year | Modelling approach, MD groups | Total (internalising and externalising) | Internalising behaviours | Externalising behaviours | Social competence | Covariates adjusted in the final model |
|---|---|---|---|---|---|---|
| Cents et al. 2013 [31] | Group-based semiparametric method; 1. no depressive symptoms (34%), 2. low depressive symptoms (54%), 3. moderate depressive symptoms (11%), 4. high depressive symptoms (1.5%) | | Unstandardised B(CI) significance level: 1 = (ref); 2 = 0.35 (0.26 to 0.45)***; 3 = 0.79 (0.66 to 0.92)***, 4 = 1.21 (0.93 to 1.49)*** | Unstandardised B(CI) significance level: 1 = (ref); 2 = 0.35(0.24 to 0.54)***; 3 = 0.73(0.58 to 0.88)***, 4 = 0.99 (0.68 to 1.30)*** | | Maternal age, educational level, family income, ethnicity, parity, marital status, and history of a clinically significant depressed mood |
| Fihrer et al. 2009 [59] | Separate groups according to presence or absence of depressive symptoms; 1. depression in the postpartum year, 2. concurrent depression, 3. mediating role of concurrent depression | | Unstandardised B(SE) significance level: 1. 7.40 (2.43)**, 2. 0.47 (.015)** | Unstandardised B(SE) significance level: 1. 5.62 (2.59)*, 2. 0.53 (.015)*** | | Maternal education, non-English speaker background |
| Campbell et al. 2007 [30] | Group-based semiparametric method; 1. low–stable (45.6%), 2. moderate–stable (36.4%), 3. intermittent depression (3.6%), 4. moderate–increasing (6.2%), 5. high–decreasing (5.6%), 6. high–chronic (2.5%) | | CBCL: $F(5, 1250) = 20.79$, $p < .001$, Partial Eta squared = .08; Group comparisons: 1 < 2, 4, 5, 6*; 2, 3 < 4, 5* | CBCL: $F(5, 1250) = 10.99$, $p < .001$, Partial Eta squared = .06; Group comparisons: 1< 2, 4, 5, 6*; 2, 3 < 4** | Social competence: $F(5, 1250) = 15.22$, $p < .001$, Partial Eta squared = .04; Group comparisons: 1 > 2, 4, 5* 2 > 4, 5** | Maternal age, maternal education, ethnicity, marital stability, income |
| NICHD 1999 [63] | Separate groups according to presence or absence of depressive symptoms; 1. never depressed (54.5%), 2. sometimes depressed (37.9%), 3. chronically depressed (7.6%) | | | Behaviour problems: $F(2, 1150) = 64.63$, $P < .001$; pairwise comparisons: 3 vs 2 (d = .62), 2 vs 1 (d = .52), 3 vs 1 (d = 1.14) | Cooperation: $F(2, 1150) = 30.63$, $p < .001$; pairwise comparisons: 3 vs 2 (d = .46), 2 vs 1 (d = .34), 3 vs 1 (d = .81) | Site, maternal education, child sex, and birth order |
| Josefsson et al. 2007 [61] | Separate groups according to presence or absence of depressive symptoms; 1. no depressive symptoms, 2. postpartum depressive symptoms (only), 3. current depressive symptoms (only), 4. both postpartum and current depressive symptoms | OR(CI) significance level: 1 = (ref); 2 = non-significant; 3 = 4.71(1.88 to 11.78)***, 4 = 3.71(1.75 to 7.87)*** | | | | Child's age and gender, maternal age |

(*Continued*)

**Table 2.** (Continued)

| Author, Year | Modelling approach, MD groups | Total (internalising and externalising) | Internalising behaviours | Externalising behaviours | Social competence | Covariates adjusted in the final model |
|---|---|---|---|---|---|---|
| Trapolini et al. 2007 [64] | Separate groups according to presence or absence of depressive symptoms; 1. never depressed (23%), 2. sometimes depressed (37%), 3. chronically depressed (40%) | | $F(2, 81) = 10.64$, $P < 0.01$; (d = 0.32) | $F(2,81) = 9.04$, $P < 0.01$; (d = 0.31) | | Maternal education, non-English speaker background, and child gender |
| Luoma et al. 2001 [62] | Separate groups according to presence or absence of depressive symptoms; 1. with depressive symptoms at different time points (D), 2. without depressive symptoms at different time points (ND) | Proportions (%) of children having a high level of problems: Prenatal ND = 13; D = 56**; Postnatal ND = 16; D = 39; Current ND = 15; D = 45* | Proportions (%) of children having a high level of problems: Prenatal ND = 18; D = 31; Postnatal ND = 18; D = 39; Current ND = 19; D = 27 | Proportions (%) of children having a high level of problems: Prenatal ND = 8; D = 38**; Postnatal ND = 10; D = 15; Current ND = 10; D = 27 | | |
| Brennan et al. 2000 [35] | Separate groups according to presence or absence of depressive symptoms; 1. neither severe nor chronic, 2. chronic but not severe, 3. severe but not chronic, 4. both chronic and severe; Timing: 1. during pregnancy only, 2. at birth only, 3. at 6 months only, 4. at 5 years only | Standardised B, significance level: Severity = .26***; Chronicity = .25***; Severity X Chronicity = -.08*; Severity × Chronicity interaction $F(1,2011) = 6.53$, $p < .05$, (d = .11); Timing in moderate depressive symptoms: $F(3, 881) = 2.87$, $p < .05$, (d = .19); Timing in severe depressive symptoms: $F(3, 205) = 5.54$, $p < .01$, (d = .33) | | | | Gender and birth order of child, mother's age and education, family income, and number of changes in mother's marital status |

$^*$ = $p < .05$.

$^{**}$ = $p < .01$.

$^{***}$ = $p < .001$.

**Notes**: Table shows only the mother's report. [51] reported a composite measure between mother's and teacher's reports, and the following papers also described teachers reports that are not included in this table: [27, 56, 59, 61]. Cohens d: 0.2 = small effect size, 0.5 = medium effect size, 0.8 = large effect size. Odds Ratio (OR) effect sizes were estimated using a Cohen's d conversion table. Partial eta squared for ANOVA: 0.01 = small effect size, 0.06 = medium effect size, 0.14 = large effect size.

[OR = 6.9 to 8.3] [33, 58], medium in three studies [OR = 2.6; Cramer's V = 0.32] [30, 51, 65], and small in four studies [OR = 1.89; $\eta_p^2 = 0.25$] [54, 57, 60, 64]. This would suggest variability in the magnitude of associations with internalising outcomes, with larger effects for more chronic symptoms. Whilst significant associations with children's internalising problems were found for mothers in the chronic and more severe groups, some of these same studies failed to find statistically significant associations with internalising outcomes for mothers who reported depressive symptoms at baseline only (three months after birth) [58], had high symptoms in pregnancy only [36], decreasing symptoms following the antenatal period [57], and groups with early postpartum depressive symptoms, subclinical depressive symptoms, and high depressive symptoms [54]. Studies using teachers reports found no significant associations

with children's internalising symptoms [30, 59], with one exception, which found a small effect [d = 0.31] [62].

## Externalising behaviours

Nineteen studies examined children's externalising outcomes. Using maternal report (eighteen studies), almost all the studies showed significant associations between MD groups and externalising outcomes, apart from three studies [52, 56, 59]. In Fihrer et al. (2009) [59], the effects of early MD were fully mediated by concurrent maternal depressive symptoms. Overall, children of mothers in the low or no depressive symptoms groups displayed significantly fewer externalising problems than children of mothers with chronic and more severe depressive symptoms. Reported effect sizes were large in four studies [OR = 4.84 to 8.90] [32, 33, 58, 63], medium in three studies [OR = 2.17; $\eta_p^2$ = 0.06] [29, 52, 66] and small in four studies [d = 0.31; $\eta_p^2$ = 0.03] [57, 60, 64, 65]. Whilst significant associations with children's externalising problems were found for mothers in the chronic and more severe groups, some of these same studies failed to find statistically significant associations with externalising outcomes for mothers who reported depressive symptoms at baseline only [58], high symptoms in pregnancy only [36], early postpartum depressive symptoms [54], high symptoms in children's preschool period only [36], increasing and decreasing symptoms at age 3 [56]. Conversely, when using teacher reports, no significant associations with externalising outcomes were found [30, 59, 62, 64].

## Social competence

Four studies examined children's social competence. Using maternal report (four studies), all studies showed significant associations between MD groups and social competence skills such as cooperation, peer relationships and prosocial behaviour [21, 30, 36, 63]. Women with chronic or more severe depressive symptoms rated their children as less cooperative, having greater peer problems, and with low levels of prosocial behaviour and social skills as compared to mothers with lower or no depressive symptoms. Effect sizes were large in two studies [OR = 4.59; d = 0.81] [21, 63] and small in another [$\eta_p^2$ = 0.04] [30]. However, persistent intermediate-level depressive symptoms and high symptoms in pregnancy only were not associated with maternal rated peer problems or prosocial behaviour [36]. Only one study used teacher reports and classroom observations for social skills, finding no significant association [30].

## Timing of depressive symptoms

Ten studies identified typologies according to the timing of depressive symptoms. Two studies [60, 67] identified significant associations of MD in the antenatal period with offspring internalising problems, and two studies with externalising outcomes [66, 67]. In contrast, two studies reported no associations between groups of MD in the antenatal period only with children's internalising, externalising or social competence outcomes [35, 36]. Some studies considering depressive symptoms in the postnatal period did not find significant associations with internalising, externalising or social competence outcomes [52, 58, 61], whereas other studies found significant associations with externalising problems [35, 54, 60] reporting small effect sizes [OR = 1.84; d = 0.33], and internalising outcomes [54, 60] with medium effect sizes [OR = 2.16]. Finally, concurrent depressive symptoms were associated with internalising, externalising, and social competence problems, with large and medium effect sizes [OR = 3.81 to 4.71] [36, 58, 60, 61, 67].

## Chronicity of depressive symptoms

Overall, chronic maternal depressive symptoms were associated with poorer offspring outcomes. Specifically, children of mothers with persistent depressive symptoms were associated with having higher internalising and externalising outcomes [31–33, 35, 36, 53, 56–58, 60, 61, 63–65], and poorer social competence [21, 63] than peers with non-chronically depressed mothers. These results remained significant even when the severity of symptoms was low [31, 32], or high-decreasing over time [30]. Studies reported large [OR = 4.84 to 8.8] [32, 33, 35, 58, 63], medium [OR = 2.0 to 3.71] [53, 61, 65], and small [$\eta_p^2$ = 0.02; d = 0.31] [57, 64] effect sizes. Only two studies failed to find significant associations between chronic decreasing [56] and high chronic MD trajectory groups [52] and children's outcomes.

## Severity of depressive symptoms

As expected, there was a pattern of association between chronic MD and more severe depressive symptoms across follow-ups, suggesting chronicity and severity are highly correlated [35, 59]. Mothers with persistent and severe depressive symptoms reported increased internalising, externalising outcomes and lower social competence in their children [21, 32, 33, 35, 36, 51, 53, 64, 66]; revealing large [OR = 4.84 to 8.8] [32, 33, 35], medium [OR = 2.0 to 2.05] [21, 53], and small [d = 0.31] [64] effects sizes. Results highlight that more severe symptoms are associated with larger effect sizes (i.e., increased odds ratios or standardised beta for severe compared to moderate symptoms). Furthermore, MD at low, moderate, or marked severity levels were associated with increased childhood internalising and externalising problems as compared to children of women with no depressive symptoms, with poorer childhood outcomes associated with mothers in the more severe groups [30–32, 56]. This may suggest a dose-response association. Moreover, increasing depressive symptoms were associated with internalising and externalising problems whereas, decreasing MD symptoms following the antenatal period were not associated with children's outcomes [56]. Finally, some studies did not find significant associations between the severity of symptoms and externalising [52, 54] or internalising [52, 57] problems.

## Quality assessment

A summary of methodological quality assessments is presented in Table 3. The main methodological issue across studies was the reporting and justification of sample size, where no studies reported a power analysis. However, the utility of power calculations in observational studies has been questioned [68], and power can be inferred from other sources such as confidence intervals. In addition, there were issues concerning blinding, where twenty-one of the studies did not report full or partial blinding strategies in the assessment stages. The remaining three studies reported blind coders for children's outcomes and information on the mother's depression status or family characteristics [30, 59, 65]. Moreover, there were differences regarding the types of informants used for MD and offspring measurements. In relation to MD, only three studies [59, 64, 65] used both self-report and clinical interviews to assess maternal depressive symptoms. The rest of the studies used self-report only. Nevertheless, self-report screening tools have been well validated within the literature and offer advantages beyond the classification of presence or absence of diagnosis, as is commonly obtained with clinical assessments. For children's outcomes, sixteen studies used only one type of informant [21, 32, 35, 36, 52–56, 59–61, 63, 65]. The remaining studies combined two [31, 33, 51, 57, 62] and three [30, 59, 64] types of informants, which is advantageous as it offers multiple perspectives of children's outcomes across settings. Variation in statistical analyses of MD groups/profiles were observed whereby fourteen studies used robust statistical analyses such as latent class or profile

**Table 3. Quality assessment.**

| Author, Year | Unbiased selection of cohort | Selection minimised baseline difference | Sample size calculated and number | Adequate description of the cohort | Validated method for ascertaining maternal depression | Validated method for ascertaining offspring outcomes | Outcome assessment blind to exposure |
|---|---|---|---|---|---|---|---|
| Pietikainen et al. 2020 [57] | Y | N/A | P | Y | P | Y | C/T |
| Oh et al. 2020 [55] | Y | N/A | P | Y | P | P | C/T |
| Hentges et al. 2020 [60] | | | P | Y | P | P | C/T |
| Garman et al. 2019 [52] | Y | N/A | P | Y | P | P | C/T |
| Maruyama et al. 2019 [21] | Y | N/A | P | Y | P | P | C/T |
| Gjerde et al. 2017 [67] | P | N/A | P | P | P | P | C/T |
| Wolford et al. 2017 [66] | Y | N/A | P | Y | P | P | C/T |
| Matijasevich et al. 2015 [33] | Y | N/A | P | Y | P | Y | C/T |
| Ahun et al. 2018 [51] | P | N/A | P | Y | P | Y | C/T |
| Kingston et al. 2018 [54] | Y | N/A | P | Y | P | P | C/T |
| Park et al. 2018 [56] | Y | N/A | P | Y | P | P | C/T |
| Netsi et al. 2018 [32] | P | N/A | P | P | P | P | C/T |
| Vakrat et al. 2018 [65] | Y | N | P | Y | Y | Y | Y |
| Giallo et al. 2015 [53] | Y | N/A | P | Y | P | P | C/T |
| Van Der Waerden et al. 2015 [36] | Y | N/A | P | Y | P | P | C/T |
| Agnafors et al. 2013 [58] | P | N/A | P | N | P | P | C/T |
| Cents et al. 2013 [31] | Y | N/A | P | Y | P | Y | C/T |
| Fihrer et al. 2009 [59] | Y | N/A | P | P | Y | Y | Y |
| Campbell et al. 2007 [30] | Y | N/A | P | Y | P | Y | Y |
| NICHD 1999 [63] | Y | N/A | P | Y | P | P | P |
| Josefsson et al. 2007 [61] | P | Y | P | P | P | P | C/T |
| Trapolini et al. 2007 [64] | Y | N/A | P | Y | Y | Y | C/T |
| Luoma et al. 2001 [62] | P | N/A | P | P | P | Y | C/T |
| Brennan et al. 2000 [35] | Y | N/A | P | Y | P | P | C/T |
| Author, Year | Adequate follow-up period | Report missing data/ drop-out | Analysis control for confounding | Analytic methods appropriate | Total score | Maximum score | Percentage score |

*(Continued)*

**Table 3.** (Continued)

| Author, Year | Unbiased selection of cohort | Selection minimised baseline difference | Sample size calculated and number | Adequate description of the cohort | Validated method for ascertaining maternal depression | Validated method for ascertaining offspring outcomes | Outcome assessment blind to exposure |
|---|---|---|---|---|---|---|---|
| Pietikainen et al. 2020 [57] | Y | P | Y | Y | 14 | 20 | 70% |
| Oh et al. 2020 [55] | Y | Y | P | Y | 14 | 20 | 70% |
| Hentges et al. 2020 [60] | Y | Y | Y | P | 14 | 20 | 70% |
| Garman et al. 2019 [52] | Y | Y | Y | Y | 15 | 20 | 75% |
| Maruyama et al. 2019 [21] | P | Y | Y | Y | 14 | 20 | 70% |
| Gjerde et al. 2017 [67] | Y | P | Y | Y | 12 | 20 | 60% |
| Wolford et al. 2017 [66] | Y | N | Y | Y | 13 | 20 | 65% |
| Matijasevich et al. 2015 [33] | Y | Y | Y | Y | 16 | 20 | 80% |
| Ahun et al. 2018 [51] | Y | Y | Y | Y | 15 | 20 | 75% |
| Kingston et al. 2018 [54] | Y | Y | Y | Y | 15 | 20 | 75% |
| Park et al. 2018 [56] | P | P | Y | Y | 13 | 20 | 65% |
| Netsi et al. 2018 [32] | P | Y | P | Y | 11 | 20 | 55% |
| Vakrat et al. 2018 [65] | P | N | N | N | 12 | 22 | 54.5% |
| Giallo et al. 2015 [53] | Y | Y | Y | Y | 15 | 20 | 75% |
| Van Der Waerden et al. 2015 [36] | Y | P | Y | Y | 14 | 20 | 70% |
| Agnafors et al. 2013 [58] | N | N | N | P | 5 | 20 | 25% |
| Cents et al. 2013 [31] | Y | Y | Y | Y | 16 | 20 | 80% |
| Fihrer et al. 2009 [59] | Y | P | P | P | 15 | 20 | 75% |
| Campbell et al. 2007 [30] | Y | Y | Y | Y | 18 | 20 | 90% |
| NICHD 1999 [63] | Y | Y | Y | P | 15 | 20 | 75% |
| Josefsson et al. 2007 [61] | N | Y | Y | P | 12 | 22 | 54.5% |
| Trapolini et al. 2007 [64] | Y | Y | Y | P | 16 | 20 | 80% |
| Luoma et al. 2001 [62] | P | N | N | N | 7 | 20 | 35% |
| Brennan et al. 2000 [35] | P | Y | Y | P | 13 | 20 | 65% |

**Notes**: Y: yes, N: no, P: partially, C/T: cannot tell, N/A: this question was not applicable for the study.

analysis, group-based modelling, and growth trajectory models [21, 30–33, 36, 51–57, 66], and ten used their own criteria to create MD groups according to the presence or absence of depressive symptoms [35, 58–65, 67]. Results were consistent between studies using robust statistical analyses and studies' own criteria to create groups. On average, there were five follow-ups after baseline measurement of MD (SD = 2.64). Only one study reported one follow-up [58], which limited its potential to assess chronicity considering the lack of longitudinal assessments. Regarding attrition rates, drop-outs were reported in all studies. Seven studies reported more than 30% attrition, and all of them included steps to minimise bias except for three [58, 62, 65]. Some studies reported attrition patterns suggesting more disadvantaged families (e.g., younger, less educated, with less financial resources mothers) had greater drop-out from follow-up than less diasvantaged families [30, 31, 35, 36, 53, 54, 56, 60, 63, 66]. The rest of the quality items reported less variability, with most papers using an unbiased selection and description of the cohort, and assessment of confounding variables.

## Discussion

We aimed to synthesise current evidence on the associations between different presentations of MD from the perinatal period onwards and offspring internalising, externalising and social competence outcomes, whilst addressing the associations of timing, chronicity, and severity of MD with these outcomes. Taken together, our findings suggest 1) substantial variability in presentations of MD from the perinatal period onwards; 2) poorer outcomes (i.e., internalising, externalising and social competence) for children exposed to MD; and that these outcomes are 3) dependent largely upon the chronicity of MD, but also symptom severity; 4) differences between informants' ratings of children's outcomes were also observed. This study extends prior work examining MD and its implications in offspring internalising, externalising, and social competence outcomes [8–10, 19, 69] by considering the complexity of different presentations of maternal depressive symptoms.

All studies described different typologies of MD, identifying between two and seven distinct MD groups. This confirms existing evidence identifying different types of MD presentations at different periods [5, 40, 70–72]. However, methodological inconsistencies between studies, particularly variation in the approach to assessment of MD groups may have produced substantial differences in the reported MD groups, and consequently hampers comparisons between studies. That said, identification of both a no/low depressive symptom group and a high chronic depressive symptom group was a common feature of the majority of identified studies. Most women were in the low-risk groups (63.9% of mothers on average), and a small proportion of mothers were in the high-risk groups (6.1% of mothers on average). Our results mirror existing systematic reviews of perinatal depressive symptoms [70, 71] demonstrating low- and high-risk groups related to the severity and persistence of maternal depressive symptoms. In addition, the majority of studies also identified transient groups such as increasing and decreasing symptoms, moderate or subclinical symptoms, and episodic depressive symptoms at some periods only. These groups are consistent with findings supporting a recovery group of mothers within the perinatal period [26], and a group at risk for future depressive symptoms, reporting symptoms up to three years after birth [26, 73]. Transient symptom groups may reflect the emergence of further challenges beyond the perinatal period. For example, transient increasing symptoms groups may reflect the impact of children's behavioural or emotional problems on maternal depressive status, with studies reporting bidirectional associations between maternal depressive symptoms and children's behavioural problems [74]. In contrast, it was noted that women with transient decreasing symptoms tend to have lower depression levels by the time their children start school [30, 56], perhaps indicative of the social

support role of school. Future research could unpack the differential impact of social determinants such as school setting on depressive symptoms across motherhood.

Our results highlight several statistically significant and practical associations between different types of MD groups and offspring internalising, externalising and social competence outcomes. For maternal report, the majority of studies concluded that high-risk groups, with chronic and severe levels of maternal depressive symptoms, are associated with higher levels of children's internalising, externalising and social competence problems, in comparison to children of mothers in low-risk groups. These results are in line with previous findings examining the negative effect of perinatal MD on children's developmental outcomes [12, 19, 38, 75, 76]. For internalising outcomes, effect sizes were mostly medium size, whereas for externalising and social competence outcomes, reported effect sizes were large. However, in the absence of meta-analysis, precise estimates of the effect of depression on outcomes are still lacking. On the other hand, our findings also provide evidence for a lack of specificity in children's socio-emotional outcomes after maternal depression exposure. This may reflect the effect of one underlying psychopathology construct [75] or dependency across socio-emotional problems in childhood [77]. However, because cited studies did not examine overlapping presentations, questions about potential independent associations with externalising or internalising outcomes remain. Thus, future research should further explore if these associations represent overlapping symptom presentations or independent associations.

Evidence for associations was less clear when teacher reports were used to measure children's outcomes. Only one in three of the studies using teacher reports found significant results, albeit with small effect sizes for internalising outcomes, with one of them using a composite measure between teachers and mothers' reports. None of them showed significant associations with externalising problems or social competence. Rater differences have also been detected in previous reviews of prenatal stress [69]. Several explanations may underpin this discrepancy between informants. First, maternal negative cognitions due to depressive symptoms may affect appraisals of one's children's behaviours and potentially over-report child problems [78]. As mothers are source of information on both the exposure (i.e., maternal depression) and the outcome (i.e., child problems), associations may be inflated due to shared method variance [79]. Specifically, in studies using the same informant for the exposure and outcome, associations between maternal depression and childhood problems may be explained by the characteristics of the informant (e.g., negative cognitions) rather than the depression exposure itself. For example, a study by Ringoot and colleagues [80] reported similar associations between maternal depression and childhood externalising and internalising problems using different informants, although the associations registered by parents with depressive symptoms were described as inflated. However, informant discrepancies may also reflect the impact of the school environment on children's behaviours [81] and teachers' preconceptions regarding children's behaviours, further linked to broader social determinants of health [82]. Discrepancies may also reflect methodological constraints since most of the studies using teacher reports included smaller sample sizes (e.g., [30, 59, 62]) than studies using maternal information, impacting upon statistical power to detect significant associations.

Findings were mostly consistent across different regions, except for one study [52] conducted in South Africa. Garman and colleagues [52] reported similar MD groups and prevalence rates than the other studies, including trajectories of chronic low, late postpartum, early postpartum, and chronic high. In contrast, results were not significant when inspecting MD group differences in children's externalising and internalising outcomes at three years old. A potential explanation is that the associations between MD and children's socio-emotional problems may differ in low-middle income countries (LMICs), where socioeconomic factors potentially play an important role. However, other studies conducted in LMICs included in

this review [21, 33] reported significant associations, although they were conducted in another continent where further cultural factors may be implicated in the association. Thus, further research inspecting underlying mechanisms between MD and children's socio-emotional outcomes is needed in LMICs, where associations may differ from high-income countries (HICs) given contexts of increased risk factors [25].

## Timing, chronicity, and severity of maternal depressive symptoms

Concerning the timing of MD, findings showed two studies describing significant associations between high symptoms during pregnancy only and offspring internalising problems, and two studies with externalising outcomes, whereas two studies described no associations with children's internalising, externalising and social competence outcomes. These results were mostly inconsistent with the argument that antenatal MD represents an independent risk factor for internalising and externalising outcomes [12, 14, 22]. This finding may suggest that the impact of antenatal depressive symptoms is weaker than the postnatal effect on children's outcomes, indicating that pregnancy-specific effects are insufficient to explain childhood internalising, externalising and social-competence problems. Thus, other mechanisms of transmission must also at least partly explain the MD and children's socio-emotional problems associations, such as genetic transmission, exposure to the mother's behaviours, and contextual stressors. However, there was selective attrition across follow-up in these studies–with women with depressive symptoms during pregnancy more likely to drop out from studies, impacting upon potential study power and underestimation of antenatal MD with its associated outcomes. Further, as only five studies report antenatal MD results and selective attrition across follow-ups, there is at present, insufficient evidence to evaluate the associations of antenatal MD on children's internalising, externalising and social competence outcomes. In the postnatal period, evidence is also inconsistent, with small effect sizes when significant associations are found. In a previous review, Grace, Evindar, and Stewart [83] identified inconsistent results in relation to the association between postnatal depression and externalising problems, suggesting the timing of depression in the postnatal period was not enough to account for children's externalising outcomes, and that other factors such as chronic exposure may better explain these associations [37, 83]. However, postnatal MD had been shown to have an independent effect on internalising behaviours [14, 38, 84], where selected studies reported medium effect sizes. Finally, concerning concurrent depressive symptoms, the majority of studies reported a significant association with internalising, externalising and social competence, reporting large to medium effect sizes. Taken together, we did not find strong evidence in support of onset of MD playing a large role in the association with children's outcomes. Existing MD at the time of assessment of children's outcomes does however play an important role.

Regarding the chronicity of depressive symptoms over time, there was consensus across studies that chronic MD was associated with poorer offspring internalising and externalising outcomes and poorer social competence, regardless of symptom severity, with consistently large effects. This is in line with previous research suggesting that mothers with chronic depressive symptoms have offspring who fare worse in both the short and long term, independent of symptom severity [70, 85]. Chronicity of maternal depressive symptoms rather than timing is likely to be crucial in children's development [37].

For severity of depressive symptoms, most studies reported that mothers who were chronically depressed had more severe symptoms. This may be a function of a higher genetic predisposition for depression [37], or exposure to additional risk factors that contribute to the severity and recurrence of depressive symptoms [43–45]. Moreover, children of mothers with persistent, more severe depressive symptoms had elevated risk for adverse internalising,

externalising and social competence outcomes in comparison to other types of MD groups, consistent with previous studies stating that chronicity and severity of depression are the strongest predictors of adverse developmental outcomes in childhood [26, 40, 86]. These associations reported mostly large and medium effect sizes. However, some studies reported no significant associations when symptoms were increasing or decreasing over time, showing that the severity of symptoms itself, particularly when transient, is a limited factor to explain internalising, externalising and social competence outcomes [42].

Despite the variability in the number of MD groups found in the included papers, there is a consensus that children of mothers who are chronically depressed over time consistently reported poorer internalising, externalising and social competence outcomes, independent of symptom severity and covariates. Moreover, chronicity and severity of symptoms are highly correlated with each other, and mothers who displayed both risk factors were also those who reported children with the highest problems. Therefore, chronicity is likely to be a clear signal for increased risk, but this can further be informed by consideration of symptom severity. These associations may reflect both environmental (e.g., maternal sensitivity) and genetic (e.g., genetic confounding) transmission of risk. For example, Gjerde and colleagues [67] reported that associations between maternal depressive symptoms and children's behavioural and emotional outcomes were explained through both shared genetic liability and direct environmental exposure, although with different relative importance throughout childhood. Future studies using modern genetic techniques such as trio designs may help to clarify the genetic contribution of maternal depressive symptoms at different stages in childhood. Nevertheless, when teacher's reports are considered, only one study showed significant associations between MD and internalising outcomes (with small effects), and none of them reported significant associations with externalising and social competence outcomes. Thus, general conclusions about MD and its associations with children's outcomes may be biased potentially due to shared method variance if they only consider maternal reports and do not incorporate other informants (such as teacher reports) and they should not be seen as evidence that MD causes poor children socio-emotional outcomes. However, meta-analytic evidence suggests that treatment of depressive symptoms was associated with both decreases in MD and improvements in child mental health [87].

In summary, the studies included in this review highlight that the classic conceptualisation of MD as a homogeneous construct is problematic and warrants revisiting. Maternal depression starting in the perinatal period and persisting beyond this phase is a complex process and women can have multiple presentations across time. Nevertheless, identification of a group of mothers exhibiting chronic high symptoms was consistent across studies, and across different regions with distinct economic structures, suggesting a global at-risk population of women whose children may be at greater risk for poorer internalising, externalising, and social competence outcomes. However, although substantial evidence from HICs are available, little research has been conducted in LMICs with inconsistent findings in these few studies in relation to the associations between MD and children's socio-emotional outcomes, demanding further inspection. The results of this review invite reflection on how to best detect specific features of each MD typology, targets for intervening early, and the impact of different typologies of MD on children's developmental outcomes.

## Implications of study quality

The methodological quality was relatively strong across studies, however there are several methodological limitations. There is no "gold standard" measurement of MD and its severity, leading to dissimilarities in the operationalisation of MD and symptom severity. Moreover, as

the selected studies used different cut-off scores, it is challenging to compare MD groups due to the inconsistent criteria across studies.

A further limitation is methodological variety in the assessment of MD groups. The most common approach to report MD groups was to separate categories according to symptom chronicity and/or severity criteria. However, these methods may oversimplify the complex process in the ontogenesis of perinatal depression. Alternatives, such as variable-centered (latent class and growth mixture modelling) or person-centered (group-based modeling) approaches to model different trajectories over time, may be more robust statistical methods to state differences between groups of MD. In addition, ten studies reported a pattern of attrition where more disadvantaged families were underrepresented. Thus, results should be interpreted carefully when generalising a broader population. Finally, there was no clear relation between the quality of studies and effect sizes. The three studies with higher quality percentages reported large, medium, and small effect sizes, including a study using teachers' reports, whilst the study with the lowest quality percentage also reported large effect sizes for internalising, externalising, and chronicity of symptoms.

## Limitations

Although the review applied rigorous eligibility and exclusion criteria, we acknowledge several limitations. First, our inclusion criteria included children from three years onwards only; several otherwise eligible studies on the topic were excluded based on this criterion [e.g., 12, 88]. However, the rationale for targeting children from the preschool period onwards reflected the challenges in detecting internalising problems in infants or toddlers, arising due to less developed verbal skills, and social competence can be difficult to assess when children have fewer opportunities to engage with peers [89]. Therefore, even if included studies reported measurement data on previous ages, we did not report these. Second, the quality assessment of the selected studies was conducted mainly by the first author, with only a proportion being reviewed by an independent researcher. Whilst this is standard practice, it may represent a potential bias in the quality appraisal of studies, despite the evaluations between raters having had good agreement. Finally, most of the included studies relied on maternal reports of both maternal depression and children's socio-emotional problems. Accordingly, due to potential overestimation of associations attributable to shared method variance, the findings should be interpreted with caution.

## Implications for research and practice

We highlight a number of implications for research, policy and practice. Further research on MD should continue assessing different presentations through a variable-centred or person-centred approach, allowing group allocation according to patterns within the population. These methods will help to better describe the course of maternal depressive symptoms, identifying different progressions that may not be identifiable using a priori categorisations. Studies using their own criteria or assignment rules to create MD groups have several limitations like assuming a priori the existence of different groups, missing the nuances of change over time (e.g., patterns of decreasing and increasing symptoms may be overlooked), and potentially confounding chronicity and severity. Therefore, looking at longitudinal change patterns instead of assumed categories may help to further disentangle symptom features such as chronicity, severity, and timing.

Moreover, to further understand trajectories of antenatal and postnatal MD, longitudinal or data linkage methodologies are required to model associations of MD prior to pregnancy. Furthermore, future research could benefit from inspecting potential mechanisms (i.e., genetics,

environmental transmission, gene-environment correlation) to explain these associations. In relation to children's assessment, we emphasise the importance of utilising different types of informants (e.g., teachers, children's own perspectives, external examinators) since the choice of the informant may impact the results estimates. Further research incorporating reporters blinded to the MD group to assess children's development would also be welcome. Future research could also incorporate paternal depression different presentations over time, given their contribution to offspring internalising and externalising outcomes [90], including the potential interaction between mothers' and fathers' depressive symptomatology (e.g., the influence of the mother/father symptoms on the other partner's wellbeing) or other challenges such as assortative mating [91].

Incorporating different presentations of MD with consideration of timing, chronicity, and severity of MD as relevant risk factors in children's development has global public health implications. Considering study findings suggesting a global at-risk population of women whose children may be at greater risk for poorer socio-emotional outcomes, public health policies should target different presentations of MD, taking into account early detection, access to suitable care, and regular follow-up. Regarding early detection, public health policies should encourage MD screenings into routine primary health care at different time points of motherhood, starting in pregnancy during routine prenatal visits, and continuing with later developmental stages during children's routine medical appointments. For successful early identification in routine health visits, it is important to strengthen mental health training of all health practitioners, and address stigma related to mental health that may obstruct the integration of mental health problems with non-specialists [24]. Moreover, identification of different presentations of MD could enable better targeting of effective evidence-based intervention resources according to the needs of differing maternal groups, with potential impact on both maternal and child health via early intervention paradigms.

## Supporting information

**S1 Checklist. PRISMA checklist.**
(PDF)

**S1 Text. Search results on February 07th, 2022.**
(PDF)

**S2 Text. Data extraction pilot.**
(PDF)

## Author Contributions

**Conceptualization:** María Francisca Morales, Lisa-Christine Girard, Angus MacBeth.

**Data curation:** María Francisca Morales, Aigli Raouna.

**Formal analysis:** María Francisca Morales.

**Investigation:** María Francisca Morales.

**Methodology:** María Francisca Morales.

**Project administration:** María Francisca Morales.

**Supervision:** Lisa-Christine Girard, Angus MacBeth.

**Writing – original draft:** María Francisca Morales.

**Writing – review & editing:** María Francisca Morales, Lisa-Christine Girard, Angus MacBeth.

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
