## [Decision Letter · Decision Letter 0]

13 Jul 2022

PGPH-D-22-00343

The association of different presentations of maternal depression with children's socio-emotional development: A systematic review

Dear Dr. Morales,

Thank you for submitting your manuscript to PLOS Global Public Health. After careful consideration, we feel that it has merit but does not fully meet PLOS Global Public Health’s publication criteria as it currently stands. Therefore, we invite you to submit a revised version of the manuscript that addresses the points raised during the review process.

We look forward to receiving your revised manuscript.

Kind regards,

Julia Robinson

Executive Editor

Journal Requirements:

1. Please note that your Data Availability Statement is currently missing the direct links to access each database. If your manuscript is accepted for publication, you will be asked to provide these details on a very short timeline. We therefore suggest that you provide this information now, though we will not hold up the peer review process if you are unable.

Additional Editor Comments (if provided):

Reviewers' comments:

Reviewer's Responses to Questions

**Comments to the Author**

1. Does this manuscript meet PLOS Global Public Health’s publication criteria? Is the manuscript technically sound, and do the data support the conclusions? The manuscript must describe methodologically and ethically rigorous research with conclusions that are appropriately drawn based on the data presented.

Reviewer #1: Yes

Reviewer #2: Partly

2. Has the statistical analysis been performed appropriately and rigorously?

Reviewer #1: N/A

Reviewer #2: N/A

3. Have the authors made all data underlying the findings in their manuscript fully available (please refer to the Data Availability Statement at the start of the manuscript PDF file)?

Reviewer #1: Yes

Reviewer #2: Yes

4. Is the manuscript presented in an intelligible fashion and written in standard English?

Reviewer #1: Yes

Reviewer #2: Yes

5. Review Comments to the Author

Reviewer #1: Morales et al. present a systematic review of the relation of maternal depressive symptoms with offspring socio-emotional development. The focus is on the temporality, the chronicity and severity of symptoms, outcomes reviewed include internalizing symptoms, externalizing symptoms and social competence. They find little evidence for an association of antenatal symptoms with offspring problems and ample evidence that chronic and more severe symptoms are related to offspring problems. These insights are not particularly novel, but the review is very careful and nicely shows both methodological problems and future directions.

Data extraction and scoring were done meticulously. I compliment the authors.

I use MD and maternal depressive symptoms interchangeable in my review.

I have several suggestions but no major concerns. I encourage the reviewers to be a bit more critical and incorporate or discuss my concerns in a possible revision.

1) the term heterogeneity is used in reviews to describe studies that cannot easily be meta-analyzed and the term is used in studies of symptom heterogeneity, for example if different symptom patterns exist. Both is not addressed; thus, I recommend to avoid the term.

2) My main concern is the very timid discussion of shared or common method variance bias. In other words, many researchers now argue that we cannot use maternal information on depression and the outcome, as relying on the same rater will systematically and substantially bias=inflate associations. Indeed, different informant provide different views on the outcome (setting or relation) but cannot be compared or contrasted, if one of the informants also reports on the exposure.

A very nice study on shared method variance bias on prenatal depression is from Ringoot et al. (admittedly the reviewer's own group):

Ringoot, Ank P., Henning Tiemeier, Vincent WV Jaddoe, Pety So, Albert Hofman, Frank C. Verhulst, and Pauline W. Jansen. "Parental depression and child well-being: young children's self-reports helped addressing biases in parent reports." Journal of clinical epidemiology 68, no. 8 (2015): 928-938.

This study suggests that maternal outcome reports in this situation inflate the association by up to 50%. Thus, the discussion on page 30 and 31 must be adapted, expanded; this problem needs much more attention and the authors must be much more critical. Shared method variance bias is a major flaw in the field and several top journals, such as AJP, will not accept child studies with maternal exposure and outcome report, if the variables have a degree of subjectivity.

see also:

Madsen, Kathrine Bang, Charlotte Ulrikka Rask, Jørn Olsen, Janni Niclasen, and Carsten Obel. "Depression-related distortions in maternal reports of child behaviour problems." European Child & Adolescent Psychiatry 29, no. 3 (2020): 275-285.

3) In this context the negative studies using teacher reports should receive more, much more attention.

4) It seems the authors do not review the quality of the study base (general population) or recruitment strategies, this is important as associations in convenience samples may be biased or not generalizable.

5) The authors argue that the association of maternal depression with externalizing disorders is stronger. In the absence of meta-analytical metrics and given the clearly overlapping effect estimates this interpretation is problematic. In contrast, I think we can see this review as providing evidence that there is a lack of specificity in outcomes. I argue this is typical for research on maternal depression and if the authors fail to agree, I would at least like to see a more critical and nuanced discussion.

6) For clinical research the question to what extent the associations reflect genetic transmission of risk is important, this should be studied with modern genetic techniques and using paternal genetics/trio design. This deserves a paragraph in the discussion, many studies simply conclude there is evidence for both environmental and genetic transmission, this may not be true e.g. antenatally.

7) Please also briefly introduce the continuum concept of symptoms across severity.

8) Please discuss why latent classes and other grouping of symptoms make sense and a continuous analysis need not be preferred, discuss in the context of patterns of decrease and increase.

9) Is there really good evidence that MD is an independent risk factor for ext and int, have longitudinal studies using repeated measures and cross-lagged panel designs or so been performed?

10) The fluctuations of symptoms/transient groups should receive more attention, in particular the decrease in symptoms.

11) Was there any relation between quality of studies and effect size?

12) The authors conclude (p.31) that the association of antenatal symptoms with child outcomes is not consistent. That is true, but why not interpret this as evidence that the impact is less strong. This can also be a lead to the mechanisms of transmission. This is not black and white issue, but the authors should reconsider their interpretation.

13) Studies of concurrent depression rated by the mother should be omitted from the present study, they are not informative.

14) Power calculations in observational studies are highly debated, I disagree they should be routinely reported, power can be deducted from the CI.

Minor points

Give LMIC in full in abstract

Page 3: line 33, why "thus" why is stress automatically a risk factor for offspring development?

Page 3: what is biological sensitization across the placental barrier, I find this a vague concept

Page 3 line 47, typo: have not 'has'

How could studies disentangle severity and chronicity? Please discuss.

Page 27, line 319, I fail to understand the concept of control groups in continuous symptom score studies

Page 31, line 386, I fail to understand the conclusion about the timing of MD in relation to externalizing problems, please clarify

Page 33, limitations: Inconsistency and heterogeneity of results is no limitation but a legitimate result. You did could work to show this.

If paternal depression is suggested as future challenge to be studied, please mention the challenge of assortative mating, the influence of mothers on fathers (and vice versa); their effects are certainly not independent

Reviewer #2: The study topic is interesting and the manuscript is generally well-written.

However, the omission of several imporant studies on the addressed study topic from the systematic review markedly limit the validity of the conclusions made.

Important, large-scale cohort studies from the ALSPAC cohort, Generation R cohort, PREDO cohort, MOBA cohort and Dream Big Consortium need to be added as study references and addressed in the systematic review before the conclusions of the review are justified. For example, the ALSPAC studies, for example by O'Donnell et al. (2014) doi:10.1017/S0954579414000029 have addressed the children at multiple ages and include method-of-choice studies on the consequences of maternal perinatal depression on child internalizing and externalizing problems. The PREDO study by Lahti et al. (2017) assessed children at an average age of 3.5 years and should be included in the review. Wolford and colleagues addressed ADHD symptoms in children over 3 years of age. A recent publication from the Dream Big Consortium by Szekely et al. (2021) https://doi.org/10.1016/j.jaac.2020.02.017 have elaborately summarized study findings on the addressed topic from ALSPAC, Generation R and MAVAN cohorts. Gjerde and colleagues have in multiple publications addressed the consequences of maternal depression on child behaviour in the MOBA cohort.

Please also carefully go through the review by Van Den Bergh et al. (2020): Prenatal developmental origins of behavior and mental health: The influence of maternal stress in pregnancy and Robinson et al. (2019: https://doi.org/10.1038/s41390-018-0173-y and Madigan et al. 2018: doi: 10.1016/j.jaac.2018.06.012 for more studies on the addressed study topic.

Before all the necessary studies are included, the review is not systematic but selective. In its current format, the reviewer is uncertain what the key addition of the current review to the study literature is, given there are multiple reviews on this study topic, some of which are more comprehensive than the current one.

If exclusion has been done based on public availability of the studies, the authors could have very well at least included key results from studies where this data was not available due to data sensitivity issues.

The rationale for excluding children with age below 3 years is not very thoroughly justified, as available instruments such as CBCL do assess child externalizing and internalizing problems from 1.5 years of age onwards and these early symptoms do have predictive value for later mental health also. It also remain unclear whether average age of the child is the key criterion for exclusion and what was done in studies where the whole age range was not reported

Additionally, the definition of social competencies should be pointed out more clearly-which studies are included here.

In assessment of study quality and discrepant study findings, careful consideration of the statistical power due to differences in sample size should be taken into account. The studies using teacher reports have generally had much smaller sample sizes.

A Newcastle Ottawa Scale assessment by two independent reviewers should be conducted on the validity of the included studies, but only after all necessary studies are included in the review. I

6. PLOS authors have the option to publish the peer review history of their article (what does this mean?). If published, this will include your full peer review and any attached files.

**Do you want your identity to be public for this peer review?** For information about this choice, including consent withdrawal, please see our Privacy Policy.

Reviewer #1: **Yes: **Henning Tiemeier, Harvard T.H. Chan School of Public Health

Reviewer #2: No

---

## [Decision Letter · Decision Letter 1]

15 Sep 2022

PGPH-D-22-00343R1

The association of different presentations of maternal depression with children's socio-emotional development: A systematic review

Dear Dr. Morales,

Thank you for submitting your manuscript to PLOS Global Public Health. After careful consideration, we feel that it has merit but does not fully meet PLOS Global Public Health’s publication criteria as it currently stands. Therefore, we invite you to submit a revised version of the manuscript that addresses the points raised during the review process.

Please see remaining comments from Reviewer 1 along with the attached track changes document. Please address these comments thoroughly in the revision and the response to reviewers document. Thanks much.

We look forward to receiving your revised manuscript.

Kind regards,

Julia Robinson

Staff Editor

Journal Requirements:

Additional Editor Comments (if provided):

Reviewers' comments:

Reviewer's Responses to Questions

**Comments to the Author**

1. If the authors have adequately addressed your comments raised in a previous round of review and you feel that this manuscript is now acceptable for publication, you may indicate that here to bypass the “Comments to the Author” section, enter your conflict of interest statement in the “Confidential to Editor” section, and submit your "Accept" recommendation.

Reviewer #1: All comments have been addressed

2. Does this manuscript meet PLOS Global Public Health’s publication criteria? Is the manuscript technically sound, and do the data support the conclusions? The manuscript must describe methodologically and ethically rigorous research with conclusions that are appropriately drawn based on the data presented.

Reviewer #1: Yes

3. Has the statistical analysis been performed appropriately and rigorously?

Reviewer #1: Yes

4. Have the authors made all data underlying the findings in their manuscript fully available (please refer to the Data Availability Statement at the start of the manuscript PDF file)?

Reviewer #1: Yes

5. Is the manuscript presented in an intelligible fashion and written in standard English?

Reviewer #1: Yes

6. Review Comments to the Author

Reviewer #1: The authors have been very responsive to the reviewer's suggestions and this has improved the manuscript substantially. Occasionally they carefully and respectfully disagreed, in hindsight I find my comments not helpful in these instances, thus agree with their strategy here too. The issue of shared method variance is no prominently discussed, I am happy with the respective changes, they are thoughtful.

I really think the authors should have numbered their responses, it makes it hard to refer to them now.

I have only very minor suggestions.

1) Line 52 and point 4 of my comments addressed: in the Introduction they speak of "independent" associations with internalizing and externalizing problems. The issue of independent association is not really discussed, how do the cited studies show that internalizing and not externalizing (or vice versa) symptoms drive the association? Or do they simply mean independent of confounding, that had not need to be mentioned? The first is more challenging and interesting and poorly addressed in the manuscript. Are we measuring the association with overlapping symptom presentations or independent associations?

2) Page 27, line 319, I fail to understand the concept of control groups in continuous symptom score studies Thank you for highlighting these issues. Answer: This sentence has been modified to: “The majority of studies did not report control groups (i.e., groups of women without depressive symptoms).” (Page 28).

I do not agree with this answer, in an analysis of continuous symptoms not only the persons with no symptoms can be viewed as "control" group, this concept does not exist and the expectation that studies report the number of persons who did not endorse any symptoms is inappropriate.

3) What does it mean: "suggesting the timing of depression in the postnatal period was insufficient to account for children’s externalising outcomes"?

Insufficient exposure duration? I fail to understand.

See attachment for more ...

7. PLOS authors have the option to publish the peer review history of their article (what does this mean?). If published, this will include your full peer review and any attached files.

**Do you want your identity to be public for this peer review?** For information about this choice, including consent withdrawal, please see our Privacy Policy.

Reviewer #1: **Yes: **Henning Tiemeier

---

## [Editor Report · Decision Letter 2]

3 Feb 2023

The association of different presentations of maternal depression with children's socio-emotional development: A systematic review

PGPH-D-22-00343R2

Dear Ms Morales,

We are pleased to inform you that your manuscript 'The association of different presentations of maternal depression with children's socio-emotional development: A systematic review' has been provisionally accepted for publication in PLOS Global Public Health.

Best regards,

Julia Robinson

Executive Editor